medical physics/biocomplexity/biomathematics

actigraphy, acute insomnia, non-parametric analysis, acrophase, circadian cycle, ultradian cycles

**Author for correspondence:**
Ana Leonor Rivera
e-mail: ana.rivera@nucleares.unam.mx

# A non-parametric model: free analysis of actigraphic recordings of acute insomnia patients

Arlex Marín-García[1], Ruben Fossion[1,2],
Markus F. Müller[1,3,5], Wady Ríos-Herrera[1,4] and
Ana Leonor Rivera[1,2]

[1]Centro de Ciencias de la Complejidad (C3), Universidad Nacional Autónoma de México, CDMX, México
[2]Instituto de Ciencias Nucleares, Universidad Nacional Autónoma de México, CDMX, México
[3]Centro de Investigación en Ciencias, Universidad Autónoma del Estado de Morelos, Cuernavaca, Morelos, México
[4]Facultad de Psicología, Universidad Nacional Autónoma de México, CDMX, México
[5]Centro Internacional de Ciencias A.C., Cuernavaca, Morelos, México

RF, 0000-0001-8456-2075; MFM, 0000-0002-2653-6204; ALR, 0000-0002-0296-7966

Both parametric and non-parametric approaches to time-series analysis have advantages and drawbacks. Parametric methods, although powerful and widely used, can yield inconsistent results due to the oversimplification of the observed phenomena. They require the setting of arbitrary constants for their creation and refinement, and, although these constants relate to assumptions about the observed systems, it can lead to erroneous results when treating a very complex problem with a sizable list of unknowns. Their non-parametric counterparts, instead, are more widely applicable but present a higher detrimental sensitivity to noise and low density in the data. For the case of approximately periodic phenomena, such as human actigraphic time series, parametric methods are widely used and concepts such as *acrophase* are key in chronobiology, especially when studying healthy and diseased human populations. In this work, we present a non-parametric method of analysis of actigraphic time series from insomniac patients and healthy age-matched controls. The method is fully data-driven, reproduces previous results in the context of activity offset delay and, crucially, extends the concept of *acrophase* not only to *circadian* but also for *ultradian* spectral components.

# 1. Introduction

*Circadian*, *ultradian* and *infradian* cycles are physiological, mental and behavioural patterns of activity that exhibit approximately stable periodicity at time scales of different lengths [1–8]. A large body of evidence has been accumulated establishing a direct link between physiological systems and central and peripheral biological clocks, as well as their disturbances in disease [9]. Actigraphy, the monitoring of physical activity, is a non-invasive and ecologically valid method to study behavioural patterns in humans [10–19], even when it has some limitations [19–21]. Through the study of actigraphic recordings it has been observed that perturbations from the healthy condition of circadian rhythms are present in a number of illnesses, both physiological [22] and psychological [10–15]. Specifically, studies using actigraphy to determine in which manner circadian and ultradian disruptions are a general feature of insomnia offer mixed results [3]. Even when a wide variety of parametric methods for the analysis of actigraphic recordings exist [23–28], non-parametric, model-free analyses have shown a greater sensitivity to subtle differences between health and disease [28–37].

Acrophase, a key concept in the study of circadian cycles stemming from cosinor analysis [26], relates to the notion of phase in oscillatory components that make up the approximately periodic observed time series. It consists of the phase of a single sinusoidal component which better fits to the empirical data, and it has been extensively used to differentiate between healthy and diseased groups. Crucially, it can be extended to include an increased number of oscillatory components [23,38–42], and though this extension put the generated cosinor model closer to Fourier analyses of actigraphy. The obtained *acrophases* are difficult to interpret clinically as they do not correspond directly to actual observable behavioural patterns for every included oscillatory component. The addition of harmonics to the single cosinor model does improves the fit in general, but to what extent this arbitrary inclusion of harmonic components is a valid reflection of actual biological oscillations remains unclear.

In this context, and given the multioscillatory nature of physiological [2,43–49] and psychological [1,50–53] cycles at the molecular, tissue, and system levels, it would be reasonable to extend the concept of acrophase to specific spectral components or intervals which have been shown to have physiological or psychological relevance. Classically, acrophase has been only applied to circadian ($\approx$ 24 h) cycles, whereas ultradian (greater than 24 h) and infradian (less than 24 h) cycles are not subject to characterization using this concept. In order to avoid the difficulties discerning between noise and true signal in a multioscillatory parametric model of actigraphic recordings, non-parametric analyses work by describing the oscillatory components in terms of the Fourier or wavelet spectral decomposition directly. In this manner, spectral limits are solely defined by the properties of the observations without the need for the fitting of any parameter or by the arbitrary inclusion of harmonics. This approach, although studying only the Fourier amplitudes of an increased number of oscillatory components, has been applied previously [20,23,24,26,30,54–58], with the crucial distinction that Fourier *phases* have never been considered in these analyses.

In this work, disruptions in circadian and ultradian rhythms between acute insomnia patients and healthy age-matched controls can be identified by the statistical properties of continuous actigraphic time series using a non-parametric model-free analysis. On the one hand, temporal shifts of the circadian cycle between control and insomniac groups are found using a measure of the distance between group-wise probability density functions of gross motor movement without the need for a parametric analysis. On the other hand, both Fourier amplitudes and phases show statistically significant differences between groups. Specifically, Fourier amplitudes and phases of *ultradian* spectral components, and phase differences at *circadian* and *ultradian* components. First, a statistical comparison of group-wise probability density functions (PDFs) of activity as a function of time, and the estimation of their group-wise temporal shift at 24 h time scale, opens the possibility of non-parametric statistical testing using well-defined mathematical measures of distance between PDFs for intra- and inter-group comparisons. Second, the statistical testing for equality for 24 h time scale of both the Fourier amplitudes and phases at all oscillatory components can be used to detect statistically significant changes in central values as well as in range of dispersion between groups. It is shown that the data-driven and non-parametric approach can both reproduce previous results and reveal new insights into the disturbances of circadian and ultradian rhythms in disease. Moreover, the consideration of Fourier *phases* approximating periodic time series is a novel concept that can be extended and further studied in other cyclic phenomena such as ecosystems, sunspots, climate, traffic and other complex systems.

# 2. Methodology

## 2.1. Actigraphic recordings

We performed spectral and statistical analyses of experimental actigraphic recordings for the publicly available data from previous publications [31,36]. The data collection for the original analysis was approved by the University of Glasgow Ethics Committee and was recorded with an Actiwatch device, worn at all times throughout day and night, for a period of two weeks at most and one week at least. It consists of time series of activity counts summed at $P = 1$ min epochs taken from 21 asymptomatic controls (28 ± 6 years old, 7 males and 14 females) and 18 acute insomnia subjects (25 ± 6 years old, 5 males and 13 females). In order to avoid artificial temporal shifts in activity counts due to different observation start times, all actigraphic recordings have been shifted to start at midnight. Each time series is divided into segments of ($l = 1440$) data points thus creating ensembles of time series at the 24 h time scale, allowing for individual- and group-wise comparisons. It is worth noting that although these kinds of datasets are not stationary in the broad sense of the term, they are approximately periodic, meaning that leaving extraordinary events aside, subjects exhibit stable cycles of activity throughout the observation period. This by no means excludes the presence of stochastic components in the observed signals, and it is precisely because of the ignorance about true nature of the underlying noise that non-parametric methods are preferred. Previous studies analysed actigraphic time series using the spectral approach [19,20,23,24,26,29,30,54–60]. However, in all cases, the emphasis is placed on the relevance of the *amplitudes* of each spectral component, or interval of components, disregarding the complementary information contained in the corresponding *phases*.

## 2.2. Fourier decomposition

The spectral analysis is focused in the study of amplitudes and phases of oscillatory components in Fourier space. For a discrete time series $x(t)$ of length $l$, its *(discrete) Fourier transform* reads [61]

$$\hat{y}(\omega_k) = \sum_{t=0}^{t=l-1} x(t) \cdot \exp^{-((i2\pi)/l)tk} . \tag{2.1}$$

In the case of a real-valued discrete time series, only $l/2$ spectral components have physical interpretation. This applies for both Fourier amplitudes

$$A_k^2 = \|\hat{y}(\omega_k)\|^2 = \mathrm{Re}\{\hat{y}(\omega_k)\}^2 + \mathrm{Im}\{\hat{y}(\omega_k)\}^2 \tag{2.2}$$

and Fourier phases

$$\phi_k = \arctan[\mathrm{Im}\{\hat{y}(\omega_k)\}/\mathrm{Re}\{\hat{y}(\omega_k)\}]. \tag{2.3}$$

With the chosen observation length period, the first spectral component describes the amplitude and phase of the circadian ($\approx$24 h) cycle, the slowest cycle using the observation length of 24 h. Strictly speaking, all other (faster) spectral components can be described as ultradian (greater than 24 h) components. However, it is as yet unclear which ultradian spectral components can be meaningfully ascribed to biological cycles. Given the restrictions of the experimental design and data acquisition protocols, infradian (less than 24 h) spectral components are not studied in this work since datasets are normalized to a 24 h time length for all subjects in both groups.

## 2.3. Statistics

Statistical comparisons of both amplitudes and phases corresponding to different oscillatory components are performed using the Kolmogorov–Smirnoff (KS) significance test, which estimates the probability for both samples coming from the same distribution using the *cumulative density functions* and, thus, a non-parametric test [62]. Here, the obtained $p$-value for the non-parametric KS statistic is reported, both for Fourier amplitudes and phases at each spectral component. KS is often used to test for statistical equality between small samples with good results [63]. It is sensitive to both the scale (or spread) and location (or centre), and requires no parameters to be specified [64]. These features are desirable with the problem at hand, since we wish to test for equality between

samples with ranges that vary in orders of magnitude such as Fourier amplitudes, and samples that are expected to vary in location of the accumulation point such as Fourier phases, neither of which are expected to be of Gaussian origin. Other statistical tests for the case of normality test perform much better [65], but KS does not require Gaussian data.

## 2.4. Motion probability estimates

### 2.4.1. Probability density functions

Using the time-resolved activity counts $x_s(m)$ of a subject $s$ observed $D_s$ whole days (either 7 or 14), PDFs of motion as a function of time (in minutes) at 24 h time scale for both subject- and group-wise levels are constructed. In this case, the 24 h activity counts by minute normalized by the total activity counts for that day yields a PDF given that the summation over all resulting values is 1. To obtain these PDFs, each $x_s(m)$ is first partitioned into $D_s$ 24 h segments: $y_{s,k}(t)$, with $t$ referring to each *minute* and $k$ to each *day* ($t = 1, \ldots, 1440$, and $k = 1, \ldots, D_s$). Hence, to construct all the 24 h subject-wise motion probability density functions, corresponding values at each minute are normalized with the following:

$$P_{s,k}(t) = \frac{y_{s,k}(t)}{C_{s,k}}. \tag{2.4}$$

Being that there are 1440 min in a day, the total count reads

$$C_{s,k} = \sum_{t=1}^{t=1440} y_{s,k}(t). \tag{2.5}$$

Each $P_{s,k}(t)$ is a probability density function describing the likelihood of a motion happening at each minute $t$ at the day $k$ for subject $s$. Since each PDF sums up to 1, this normalization is extended to group-wise PDFs with the averaging of each activity profile pertaining to each subject in each group. This is

$$P_C(t) = \frac{1}{N_C} \sum_{s=1}^{N_C} \sum_{k=1}^{k=D_s} \frac{P_{s,k}(t)}{D_s} \tag{2.6}$$

for $s$ in the control group, and

$$P_C(t) = \frac{1}{N_I} \sum_{s=1}^{N_I} \sum_{k=1}^{k=D_s} \frac{P_{s,k}(t)}{D_s} \tag{2.7}$$

for $s$ in the insomniac group, and with $N_I$ and $N_C$ referring to the number of subjects in the insomniac and control groups, respectively.

### 2.4.2. Jaccard distances

Using the normalization described in the previous section, a non-parametric comparison between PDFs is used to estimate the distance between them. The Jaccard distance metric is a straightforward, simple and computationally efficient method to estimate the distance between probability distributions. It is closely related to the Euclidean, Mahanalobis and other distance metrics between PDFs [66], and it offers an intuitive interpretation of distance between pairs of PDFs. Based in its geometric counterpart, the Jaccard index [67], it has been since used in the form of a metric between PDFs in artificial intelligence [68], chemistry [69], bioinformatics [70,71] and generalizations to higher dimensions and multivariate systems [72].

The *Jaccard index* [67], $\hat{J}(A, B)$, between two closed objects $A$ and $B$ is defined as the quotient of intersection and the total expanded areas:

$$\hat{J}(A, B) = \frac{A \cap B}{A \cup B}. \tag{2.8}$$

This transforms into a proper distance measure, the Jaccard distance, $J(A, B)$, via

$$J(A, B) = 1 - \frac{A \cap B}{A \cup B}, \tag{2.9}$$

with extreme values $J(A, B) = 1$ and $J(A, B) = 0$ for completely disjoint and overlapping

objects, respectively. Extended to PDFs over compact support, the *Jaccard distance* [73,74] between $P$ and $Q$ is

$$J_{P,Q} = 1 - \frac{\sum_{t=1}^{t=l} \text{argmin}[P(t), Q(t)]}{\sum_{t=1}^{t=l} \text{argmax}[P(t), Q(t)]}, \tag{2.10}$$

in which extreme values $J_{P,Q} = 1$ and $J_{P,Q} = 0$ share the exact interpretation as in the geometric context. Moreover, imposing periodic boundary conditions on the time-resolved PDFs, $P(t)$ and $Q(t)$, it can be used with a time delay, $\tau$, in order to investigate temporal shifts between subject- and group-wise PDFs:

$$J_{P,Q}(\tau) = 1 - \frac{\sum_{t=1}^{t=l} \text{argmin}[P(t), Q(t+\tau)]}{\sum_{t=1}^{t=l} \text{argmax}[P(t), Q(t+\tau)]}, \tag{2.11}$$

with $(\tau = 1, \ldots, l-1)$ as the temporal shift between PDFs.

# 3. Results

## 3.1. Motion probability comparisons

Motion PDFs are obtained through the normalization of activity counts as a function of time over the span of one day (1440 min). Figure 1*a*,*b* presents the estimated motor activity PDFs in day time scale for two typical cases corresponding to the control and insomniac groups, respectively. For the case of the healthy subject, an abrupt decay in motor activity counts precede sleep onset ($t \approx 2$ min); followed by even more abrupt increase around sunrise ($t \approx 7$ h). This behaviour is not exhibited by the insomniac subject, which presents a slow decay in activity counts starting around midnight ($t \approx 24$ h), reaching a local minimum at similar times as the control subject is waking up.

Also, all motion probability counts for all days for all subjects in the control (blue circles) and insomniac (red diamonds) groups, along with per-group mean values (solid black curves) are presented in figure 1*c*,*d*, respectively. These results show that, at the group level, daily average motion probabilities are very similar in shape, but are shifted one from the other: the control group tends to go to bed and wake up earlier than the insomniac group; and while this is true, at the interval [10–20] h both groups tend to exhibit qualitatively similar features, indicating that albeit small differences, both groups behave quite similar in at least some hours of the day. This is better observed with group-wise PDFs plotted against each other as in figure 1*e*, where it can be observed that waking up ([6–8] h) and going to bed ([22–24] h) are quite different processes between groups. Finally, presented in figure 1*f*, the Jaccard distance as a function of temporal displacement is useful to reveal the temporal shift between control and insomniac groups $\tau_{\min} \approx 90$ min. Previous investigations using cosinor analysis show that the temporal displacement between sleep onset coincides with this non-parametric estimation [36]. Moreover, the shape of the Jaccard distance curve speaks of the periodic features of both motion probabilities; within a 15 min plateau, the localized minimum (at $\tau \approx 90$ min) relates to the broad maximum ($\tau \in [10 - 18]$ h) in the sense of night-time being very different from day-time activity profiles, disregarding of group, and to the expected wide range of dispersion among motion counts at each minute.

## 3.2. Fourier amplitudes and phases

Fourier spectral analyses include the testing for statistical equality between samples for both Fourier amplitudes and phases independently, and for all oscillatory components accessible at the 24 h time scale between groups. In figure 2*a*,*b*, Fourier amplitudes are presented for all 24 h segments for all subjects in the control and insomniac groups (with mean values in solid black curves), respectively. These results show that both control and insomniac groups exhibit very similar profiles of mean and dispersion values for all spectral components, with slightly larger amplitude dispersion for the insomniac group at the slowest oscillatory components ($1 < k < 8$). These differences are even more apparent for the Fourier phases, shown in figure 2*c*,*d*, with black circles and diamonds for mean values for the control and insomniac groups, respectively, presenting a larger dispersion at all spectral

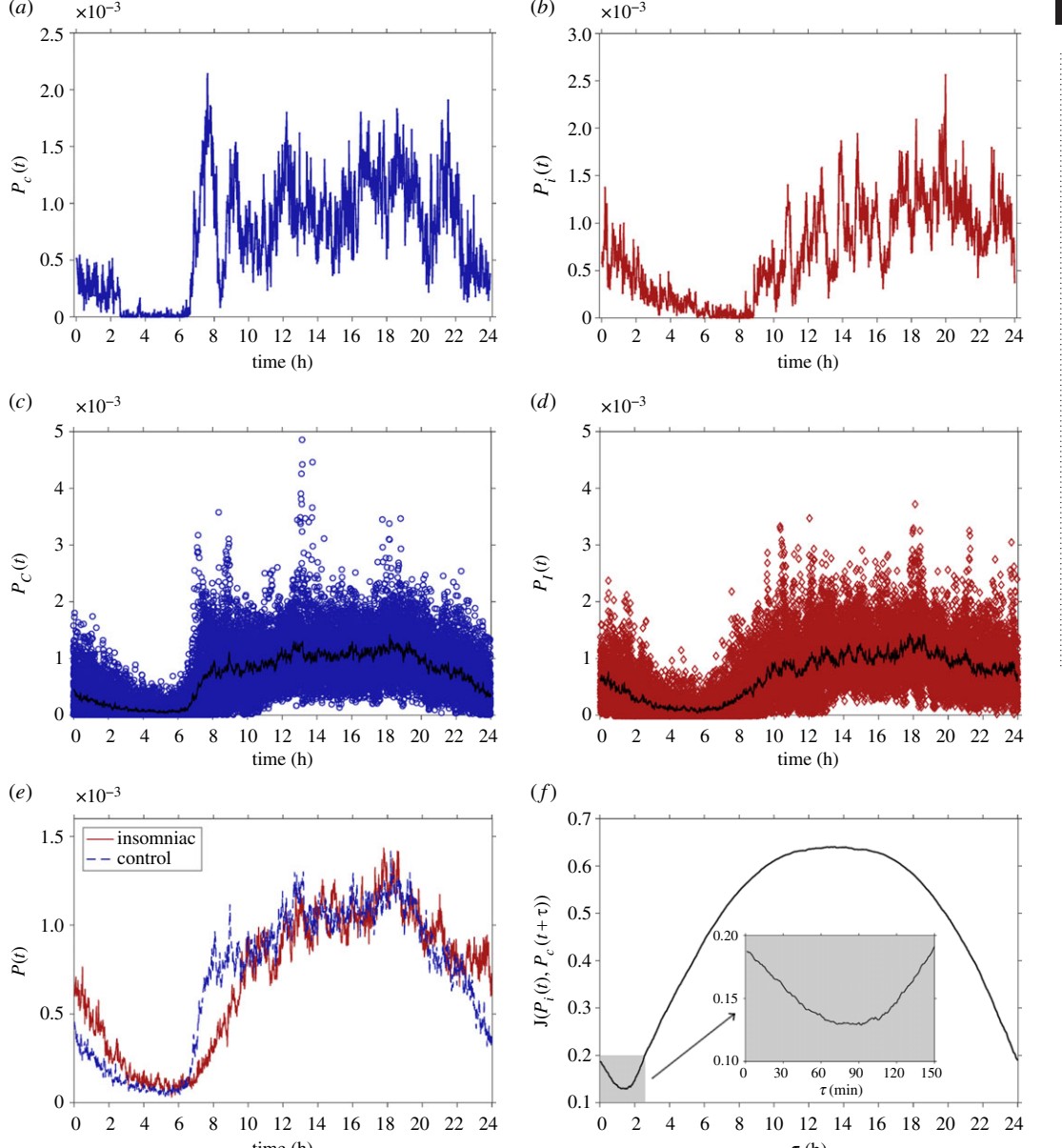

**Figure 1.** Motion probability estimates. 24 h probability of motion as a function of time for one control (a) and one insomniac subject (b). 24 h probability of motion for all control (c) and all insomniac subjects (d); mean values over all subjects and all days are plotted with solid black curves. (e) Mean values of the 24 h probability of motion for control (dashed blue curve) and insomniac (solid red curve) groups. (f) Jaccard distance between mean values of the 24 h probability of motion for control and insomniac groups as a function of temporal displacement; inset shows in detail the behaviour within the interval [0–150] min.

components, and where characteristic concentration points for the slowest oscillatory components ($1 < k < 10$) are found.

Mean values for both Fourier amplitudes (black curves in 2a,b) and phases (black symbols in figure 2c,d) for both groups are presented in figure 2e,f, respectively. In the case of Fourier amplitudes, figure 2e shows that differences in mean amplitude values between groups are larger for oscillatory components in the range ($10 < k < 30$), while figure 2f shows that larger differences in mean phase values appear consistently in the slowest components ($1 < k < 8$), with only the third component presenting no apparent difference between groups. Fourier amplitudes are very much similar in both mean value and dispersion between groups for almost all spectral components. This is not true for Fourier phases, which exhibit very distinct concentration points and dispersion at slow frequencies.

Because the experimental design, data acquisition protocols, including the length of observation periods, sampling frequency, number of participants, and unrestricted activity logging, information

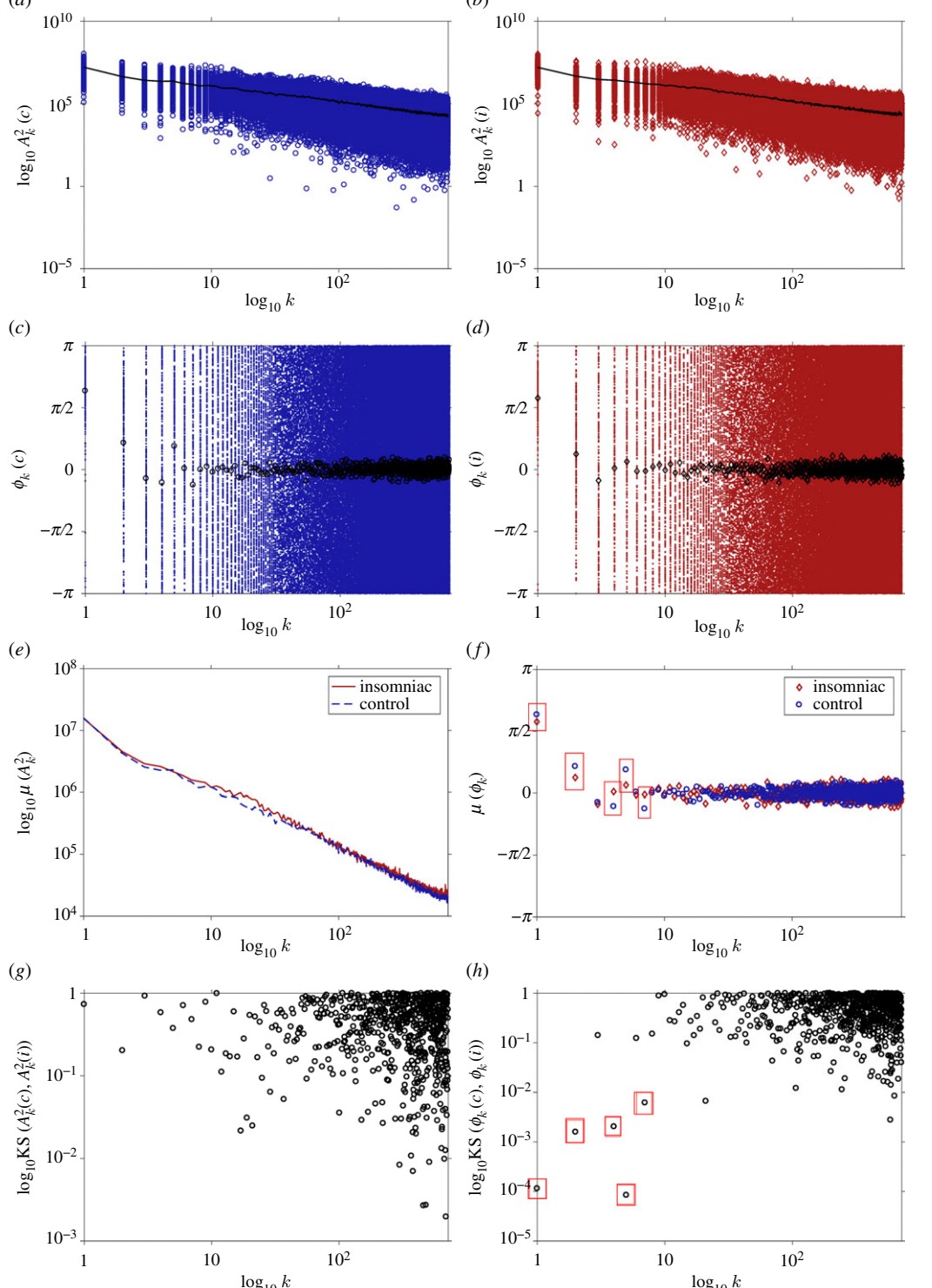

**Figure 2.** Spectral estimates. Fourier amplitudes for control (*a*) and insomniac groups (*b*) in log–log scale; mean values at each spectral component (*k*) are plotted with solid curves. Fourier phases for control (*c*) and insomniac (*d*) groups in semilog scale on the horizontal axis; mean values at each spectral component (*k*) are plotted with black circles (control group) and black diamonds (insomniac group). (*e*) Mean values of Fourier amplitudes in log–log scale for control (blue dashed curve) and insomniac (solid red curve), respectively. (*f*) Mean values of Fourier phases in semilog scale on the horizontal axis for control (blue circles) and insomniac (red diamonds), respectively. Significant ($p < 10 \times 10^{-2}$) phase differences found using the Kolmogorov–Smirnoff test are indicated by a red rectangle. (*g,h*) Results of the application of the non-parametric Kolmogorov–Smirnoff test between control and insomniac groups for Fourier amplitudes and phases at each spectral component (*k*), respectively; both panels are shown in log–log scale; significant ($p < 10 \times 10^{-2}$) phase differences in slow components found using the Kolmogorov–Smirnoff test are indicated also by a red rectangle.

regarding the range of dispersion for either activity counts and all derived measures cannot be ascribed to specific noise sources. However, since weak periodicity can be expected from a large and diverse subject pool, both central values and dispersion ranges are expected to contain useful information when establishing archetypal profiles for each group, This is true for the estimation of motion activity PDFs, as well as Fourier amplitudes and phases. Considering the fast-paced rhythm of activity observed in humans, both in healthy and diseased human populations [20,27,75], the main sources of stochasticity within the data are expected to be present at corresponding fast spectral components. Previous investigations inform the expectation of finding a activity offset delay of $\approx$ 90 min between groups [36], represented by equivalent phase difference at the first spectral component $\phi(1)_C$ and $\phi(1)_I$ for the control and insomniac groups, respectively. To support the use of these differences, tests for statistical equality between samples are first performed. Figure 2$g,h$ presents the ($\log_{10}$) $p$-values of the KS test for Fourier amplitudes and phases, respectively, and for all spectral components. Significance tests show that Fourier amplitudes show only marginal significance ($10 \times 10^{-2} < p < 10 \times 10^{-1}$) for spectral components ($k = 17$, 19, 20), whereas phases are found to be significantly different ($p < 10 \times 10^{-2}$) between control and insomniac groups in slow spectral components ($k = 1, 2, 4, 5, 7$). The difference in amplitude between groups corresponding to spectral components also called *BRAC cycles* [53] is also found through parametric methods using the same dataset [36]. In this case, for the $\approx 85$, $\approx 75$ and $\approx 72$ min periodicities (components ($k = 17$, 19, 20), respectively), significant differences in Fourier amplitudes are found as well. Notably, when transformed into minutes, phase differences between central values at the spectral components with a significant phase difference do correlate with results from previous parametric investigations, expressing a significant phase difference of 20° at the 1440 min spectral component ($\phi(1)$), which translates to 80 min when expressed in terms of that cycle. This is within the plateau found for the Jaccard distance between group-wise activity profiles in figure 1$f$. Using the available information, it is unclear as to the meaning of significance of phase differences found in other spectral components, namely $\phi(2)$, $\phi(4)$, $\phi(5)$, and $\phi(7)$, since only for the 24 h cycle ($\phi(1)$) can be intuitively related to physiological changes [76].

# 4. Discussion

This study further demonstrates that continuous actigraphic time series of insomniac and control populations can be of use for the detection of differences between healthy and diseased populations. This can be done by establishing group- and subject-wise activity profiles through normalization and averaging and the testing for statistical significance between geometric and spectral characteristics at the group level. Coupled with a fully non-parametric and data-driven methodology, it yields novel results useful for the characterization of temporal shifts between groups, and for the assessment of significance of differences both in Fourier amplitudes and phases at all spectral components, while reproducing previous results [36] and establishing a connection between phase differences in key spectral components. The current study has two main strengths. First, the use of actigraphic data allows for ecological validity [18] and thus it makes our findings more generalizable and replicable. Second, we assessed motion activity counts with non-parametric, model-free methods such that no assumption is made regarding the origins of the data and tested for statistical equality between observed samples using a non-parametric significance test that is sensitive to changes in both centrality and dispersion without selecting any particular model for either the data or the inherent noise within it.

All of this comes with limitations due to the experimental design, data acquisition protocols, observation period length, sampling density and the sizes of each group. These limitations prevented us for controlling for age, sex and socio-economic status, variables which certainly play an important role in this disease as well as in chronobiology in general. Since all actigraphic data are normalized to the 24 h time scale, no information about *infradian* components can be obtained, and with the 1 min temporal resolution, only a limited specificity of spectral components is achievable. In comparison to wavelet methods, it is worth mentioning that wavelet decompositions are not periodic, so each wavelet function involves a band of frequencies rather than a single one. In switching to Fourier analysis, with its precise frequency determination but lack of time localization, each spectral component can be determined and associated with specific physiological and environmental factors, such as light–darkness, melatonin-onset, and other cycles, physiological or psychological. Then, sample size and group size might play a role in detecting statistically

significant differences. A large sample size both in number of participants and observation time is needed to increase the significance of these findings. In this regard, although no assumption is made about noise sources and stochastic components within the data, it is reasonable to expect that increased dispersion in all observables will limit the differentiation power of any statistical test, disregarding whether parametric or not. Because of the novelty of the method and no restrictions on data logging, changes in both the central values and in their dispersion should be considered. Because of this we employ the non-parametric KS test for statistical equality, since it is sensitive to expected changes in location and spread, it uses the ranks of the observed samples instead of the actual values, and performs well even for small samples.

The geometrically inspired Jaccard distance metric can be successfully used to estimate the activity offset delay between groups, as previously reported [36], of ≈90 min. In this case, although no assumptions about noise sources are made, the wide ranges of possible values when establishing subject- and group-wise activity profiles are constrained with the normalization strategy used; with this, the distance metric provides useful information about relevant characteristics between activity profiles. It is important to note that the value is not exactly 90min, but appears within a 15 min plateau which we understand as being both related to the dispersion of values expected from a diverse population and due to independent noise sources. For this, group-wise comparisons should offer much higher statistical significance since although not small (7–14 days) observation periods of this length offer a relatively small number of observations at the minute resolution.

The proposed Fourier spectral analysis relies on the characterization of differences between spectral components which are purely determined by the properties of the observed phenomena, thus increasing the validity of our results. In contrast to other multioscillatory spectral analyses, such as multioscillatory wavelet [30] and cosinor methods [23] (based on the inclusion of harmonics to increase the model fit to observed samples), here the observed data samples provide the spectral characteristics of the *acrophases* in the Fourier sense; being a novel characterization of group-wise motion profiles, we provide a validation of this with the accounting for the observed activity offset delay between groups (≈90 min) as phase differences (expressed in minutes) for the first spectral component, corresponding to the light–dark 24 h cycle. Other components also present a significant phase differences; however, correct interpretation of these differences should be accompanied by the simultaneous study of several biological markers such as blood pressure, heart rate, skin temperature among others, in order to provide a clear association between physiological and psychological alterations [1,2,8,76]. It suggests, however, that a *cascading* regulation between environmental and physiological changes and the multioscillatory nature of biological rhythms can induce perturbations in the coupling of the sleep cycle with circadian cycle resulting in perturbations in more than just the spectral component pertaining to the circadian rhythms.

# 5. Conclusion

The use of continuous actigraphic data was valuable for establishing the profile of the temporal activity offset patterns of the evaluated populations; that spectral and statistical properties for infradian, circadian and ultradian rhythms in humans can be studied through continuous actigraphic recordings and using non-parametric approaches; that the analysis of both Fourier amplitudes and phases is helpful in finding statistically meaningful differences between insomniac and control groups; and that the use of non-parametric geometric-inspired measures of distance between PDFs concur with parametric estimations of acrophase delay between the datasets.

Ethics. We performed spectral and statistical analyses of experimental actigraphic recordings for the publicly available data from a previous publication [31]. The data collection for the original analysis was approved by the University of Glasgow Ethics Committee, as stated in the publication. Ethics Commitee from Facultad de Medicina of UNAM approved the project FM/DI152/2016 'Salud y enfermedad: un enfoque desde las Ciencias de la Complejidad' related to the analysis of non-invasive signals like actigraphic records.

Data accessibility. Dryad doi:10.5061/dryad.0k6djhb1f. Data have been published previously [31]. Prof. Jason Ellis give us permission to use the data published in that paper (enclosed in submission: Permission.pdf).

Authors' contributions. A.M.-G.: formal analysis, methodology, visualization, writing—original draft, writing—review & editing; R.F.: conceptualization, formal analysis, writing—original draft, writing—review & editing; M.F.M.: investigation, validation, writing—review & editing; W.R.-H.: data curation, formal analysis, validation,

visualization, writing—review & editing; A.L.R.: conceptualization, funding acquisition, investigation, methodology, supervision, validation, writing—original draft, writing—review & editing.

Competing interests. We declare we have no competing interests.

Funding. This work was partially financially supported by CONACyT through Fronteras grant no. FC-2016-1/2277 and through FORDECYT-PRONACES grant nos. 2020/610285 and 2020/263377; the Universidad Nacional Autónoma de México through DGAPA-PAPIIT grant nos. IV100120, IN113619, IA100522 and PAPIME grant no. PE103519. Funders had no role in the manuscript writing, editing, approval, or decision to publish.

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
