## [Peer Review File · Royal Society Open Science]

Review History

RSOS-210463.R0 (Original submission)

Review form: Reviewer 1 (David Halliday)

Is the manuscript scientifically sound in its present form?

Yes

Are the interpretations and conclusions justified by the results?

No

Is the language acceptable?

Yes

Do you have any ethical concerns with this paper?

No

Have you any concerns about statistical analyses in this paper?

No

Recommendation?

Major revision is needed (please make suggestions in comments)

Comments to the Author(s)

This manuscript considers analysis of time series actigraph data comparing recordings from control and insomniac groups using Fourier decomposition of activity counts to assess differences in the two populations. My comments fall into two areas 1) Methodology/results and 2) Interpretation.

1. Methodology. The rationale for using non parametric methods is discussed in the manuscript, the motivation given appears reasonable. In section 2 the techniques are presented, this section could benefit from better integration between the methods, particularly the assumptions being made of the data, regarding stationarity for Fourier decomposition, presence (or otherwise) of stochastic components, and the rationale for using the Jaccard method and K-S test for comparing the activity counts and frequency analyses, respectively. The results in figure 1, and equations (4) and (5) present the data as PDF values. However, these are just mean activity counts for a single subject in (4) and across each group in (5). The use of mean activity counts seems reasonable, but presenting these as PDFs is not justified here, there is no discussion of normalisation, PDF estimation or what this is a probability of? Presenting these as measures of activity counts would be more appropriate in this case. Some additional clarity on what is presented in Figure 2 would be useful. Panels a), b) show individual log periodograms, with the average periodogram overlaid. Panel e) also contains some form of average spectral estimate, but the values are different. What is the relationship between these two sets of spectral estimates? It would be useful to include recognition that these are just estimates, a clearer statement, possibly with equation, on how the averages across subject were calculated could be included. Following up the Bloomfield reference, some indication of confidence limits on these log spectra plots would be a useful addition. This would give some context for the 4-5 orders of magnitude we see in the ranges of values for individual subjects.

2. Interpretation of results is weak in the context of comparing the two groups. The discussion contains 3 numbered points: 1) Phase – this just says phase is a useful measure, but makes no attempt at inference. What interpretation should the reader give to the phase difference highlighted in Fig 2f) and that sometimes the phase is greater for insomniac and sometimes greater for control. Thus, what does a phase of $\pi/2$ signify for index $k=1$? 2) The point on statistical tests should include more details of where the significant differences were found and the point regarding the need for additional analysis should contain more details of how this can be followed up. 3) This contribution is largely speculative and makes no reference to any of the results presented. For example, what do the results in Figs 1 and 2 tell us about synchronisation of circadian rhythms or neurogenesis? This needs clarification, or possibly to be removed for some more concrete discussion. A key point in the analysis is the 90 minute delay detected using the Jaccard analysis, but this is not mentioned in the discussion. What is the significance of this delay for our understanding and comparing the two groups?

Minor

1. Abstract: “concepts such as acrophase are staple in chronobiology”. Clarify meaning of this.
2. In equation (2) the time index from be 0 to $(l-1)$ for a DFT definition.
3. In Line 128 what is “proper probability”.
4. Lines 187-189 incorrect figure references.
5. Line 261, wavelet vs Fourier these criticisms also apply to the DFT where frequency resolution depends on sequence length and any data windowing.

Review form: Reviewer 2

Is the manuscript scientifically sound in its present form?

Yes

Are the interpretations and conclusions justified by the results?

No

Is the language acceptable?

Yes

Do you have any ethical concerns with this paper?

Yes

Have you any concerns about statistical analyses in this paper?

No

Recommendation?

Reject

Comments to the Author(s)

This manuscript describes a non-parametric method of analysis of actigraphic time series from 12 insomniac patients and healthy age-matched controls, that is fully data-driven.

Detailed comments are below:

On Page 4, Line 59, the authors state that their technique avoids the difficulties of making assumptions about noise and true signal. But this means the outcome of the analyses provide no insights on whether the results reflect noise or signal. How is this resolved?

On Page 9, Line 166 the authors discuss results from two subjects to highlight differences between healthy sleepers and insomniacs. How are these findings different from that of a traditional approach?

In page 15, Line 252, the authors state ' we assessed motion activity counts with non-parametric, model-free methods such that no assumption is made regarding the origins of the data; such assumptions are particularly problematic in diseased populations with less pronounced circadian rhythms.' Why is that a problem in individuals with less pronounced circadian rhythms? Please explain with specific reasons.

On page 16, line 265 the authors state 'each spectral component can be determined and associated with specific physiological and environmental factors, such as light-darkness, melatonin-onset, and other cycles, physiological or psychological. It would be useful to highlight these associations with the results of your analyses. Also what psychological cycles are you referring to?

The authors have provided a new analytic technique for actigraphy data. It would be useful if they could substantiate with specific, explicit results their statements in the discussion and conclusions. For example in their conclusions the authors state ' that spectral and statistical properties for infradian, circadian and ultradian rhythms in human can be studied through continuous actigraphic recordings and using non-parametric approaches.' Please provide specific examples with your data analyses. Secondly, please discuss in detail the environmental cycles to which the results of your analyses can be anchored.

Decision letter (RSOS-210463.R0)

Dear Dr Rivera

The Editors assigned to your paper RSOS-210463 "A non-parametric model-free analysis of actigraphic recordings of acute insomnia patients" have now received comments from reviewers and would like you to revise the paper in accordance with the reviewer comments and any comments from the Editors. Please note this decision does not guarantee eventual acceptance.

Please submit your revised manuscript and required files (see below) no later than 21 days from today's (ie 11-Aug-2021) date. Note: the ScholarOne system will 'lock' if submission of the revision is attempted 21 or more days after the deadline. If you do not think you will be able to meet this deadline please contact the editorial office immediately.

on behalf of Professor Mark Chaplain (Subject Editor)
openscience@royalsociety.org

Associate Editor Comments to Author:

The reviewers have identified a number of major concerns that prevent us from accepting the paper at this stage. You need to carefully address their concerns before further consideration is possible - indeed, the revision will go back to the reviewers for a further assessment: if they are unsatisfied with the changes made, then we may not be able to consider the work further.

Reviewer comments to Author:

Reviewer: 1

Comments to the Author(s)

This manuscript considers analysis of time series actigraph data comparing recordings from control and insomniac groups using Fourier decomposition of activity counts to assess differences in the two populations. My comments fall into two areas 1) Methodology/results and 2) Interpretation.

1. Methodology. The rationale for using non parametric methods is discussed in the manuscript, the motivation given appears reasonable. In section 2 the techniques are presented, this section could benefit from better integration between the methods, particularly the assumptions being made of the data, regarding stationarity for Fourier decomposition, presence (or otherwise) of stochastic components, and the rationale for using the Jaccard method and K-S test for comparing the activity counts and frequency analyses, respectively. The results in figure 1, and equations (4) and (5) present the data as PDF values. However, these are just mean activity counts for a single subject in (4) and across each group in (5). The use of mean activity counts seems reasonable, but presenting these as PDFs is not justified here, there is no discussion of normalisation, PDF estimation or what this is a probability of? Presenting these as measures of activity counts would be more appropriate in this case. Some additional clarity on what is presented in Figure 2 would be useful. Panels a), b) show individual log periodograms, with the average periodogram overlaid. Panel e) also contains some form of average spectral estimate, but the values are different. What is the relationship between these two sets of spectral estimates? It would be useful to include recognition that these are just estimates, a clearer statement, possibly with equation, on how the averages across subject were calculated could be included. Following up the Bloomfield reference, some indication of confidence limits on these log spectra plots would be a useful addition. This would give some context for the 4-5 orders of magnitude we see in the ranges of values for individual subjects.

2. Interpretation of results is weak in the context of comparing the two groups. The discussion contains 3 numbered points: 1) Phase - this just says phase is a useful measure, but makes no attempt at inference. What interpretation should the reader give to the phase difference highlighted in Fig 2f) and that sometimes the phase is greater for insomniac and sometimes greater for control. Thus, what does a phase of $\pi/2$ signify for index $k=1$? 2) The point on statistical tests should include more details of where the significant differences were found and the point regarding the need for additional analysis should contain more details of how this can be followed up. 3) This contribution is largely speculative and makes no reference to any of the results presented. For example, what do the results in Figs 1 and 2 tell us about synchronisation of circadian rhythms or neurogenesis? This needs clarification, or possibly to be removed for some more concrete discussion. A key point in the analysis is the 90 minute delay detected using the Jaccard analysis, but this is not mentioned in the discussion. What is the significance of this delay for our understanding and comparing the two groups?

Minor

1. Abstract: "concepts such as acrophase are staple in chronobiology". Clarify meaning of this.
2. In equation (2) the time index from be 0 to (l-1) for a DFT definition.
3. In Line 128 what is "proper probability".
4. Lines 187-189 incorrect figure references.
5. Line 261, wavelet vs Fourier these criticisms also apply to the DFT where frequency resolution depends on sequence length and any data windowing.

Reviewer: 2

Comments to the Author(s)

This manuscript describes a non-parametric method of analysis of actigraphic time series from 12 insomniac patients and healthy age-matched controls, that is fully data-driven.

Detailed comments are below:

On Page 4, Line 59, the authors state that their technique avoids the difficulties of making assumptions about noise and true signal. But this means the outcome of the analyses provide no insights on whether the results reflect noise or signal. How is this resolved?

On Page 9, Line 166 the authors discuss results from two subjects to highlight differences between healthy sleepers and insomniacs. How are these findings different from that of a traditional approach?

In page 15, Line 252, the authors state 'we assessed motion activity counts with non-parametric, model-free methods such that no assumption is made regarding the origins of the data; such assumptions are particularly problematic in diseased populations with less pronounced circadian rhythms.' Why is that a problem in individuals with less pronounced circadian rhythms? Please explain with specific reasons.

On page 16, line 265 the authors state 'each spectral component can be determined and associated with specific physiological and environmental factors, such as light-darkness, melatonin-onset, and other cycles, physiological or psychological. It would be useful to highlight these associations with the results of your analyses. Also what psychological cycles are you referring to?

The authors have provided a new analytic technique for actigraphy data. It would be useful if they could substantiate with specific, explicit results their statements in the discussion and conclusions. For example in their conclusions the authors state 'that spectral and statistical properties for infradian, circadian and ultradian rhythms in human can be studied through continuous actigraphic recordings and using non-parametric approaches.' Please provide specific examples with your data analyses. Secondly, please discuss in detail the environmental cycles to which the results of your analyses can be anchored.

===PREPARING YOUR MANUSCRIPT===

While not essential, it will speed up the preparation of your manuscript proof if accepted if you format your references/bibliography in Vancouver style (please see

<https://royalsociety.org/journals/authors/author-guidelines/#formatting>). You should include DOIs for as many of the references as possible.

===PREPARING YOUR REVISION IN SCHOLARONE===

Author's Response to Decision Letter for (RSOS-210463.R0)

See Appendix A.

RSOS-210463.R1 (Revision)

Review form: Reviewer 2

Is the manuscript scientifically sound in its present form?

Yes

Are the interpretations and conclusions justified by the results?

Yes

Is the language acceptable?

Yes

Do you have any ethical concerns with this paper?

Yes

Have you any concerns about statistical analyses in this paper?

No

Recommendation?

Accept with minor revision (please list in comments)

Comments to the Author(s)

The authors have addressed all my comments satisfactorily.

I just have a minor comment.

It would be useful for the authors to review the manuscript for language and sentence construction. Some sentences for example lines 29-35 are too long and ones loses a sense of what tis intended. It would be good to shorten the very long sentences.

Review form: Reviewer 3

Is the manuscript scientifically sound in its present form?

Yes

Are the interpretations and conclusions justified by the results?

Yes

Is the language acceptable?

No

Do you have any ethical concerns with this paper?

No

Have you any concerns about statistical analyses in this paper?

No

Recommendation?

Accept with minor revision (please list in comments)

Comments to the Author(s)

Summary:

In the following manuscript, Marin-Garcia et al, present an alternative methodological approach to analyzing actigraphy recordings from acute insomnia patients. They compare parametric and non-parametric approaches and detail the pros and cons of each. They then present a non-parametric method to analyze these signals using a purely data-driven approach, which is meant to alleviate problems associated with adhering to increases in the number of pre-determined constants. They conclude that the “acrophase” component of these analyses can provide new insight into both circadian and ultradian phenomena.

Comments:

1. Motion probability comparisons: Figure 1 (p.16): It is unclear to me what conclusions the reader should draw from panel A and B. Furthermore, why is the data distribution not given for each group? (I realize that individual values are in panels C and D). Is it possible for the authors to also plot the standard error? Or does it make interpretation difficult due to the range of the data? Regarding the motion probability estimate, the authors states that either 7 or 14 whole days were used (p. 14, line 23). What was the reason for this? Also was any difference seen when averaging the data for these individuals? Furthermore, is this related to the number of male and female subjects in the control group?
2. Related to question one, what was the rationale for taking 1 min values? Were there any attempts to look at larger or smaller (I realize that depending on the recording device this may not be possible) time-scales? Given that the 1 min activity bout is a slightly arbitrary block of time (and may just be the smallest interval outputted by the device), do the authors think their results would change if data was summed over larger intervals as certain “discrete or stochastic” events might then contribute less?
3. The use of the term “sleep-onset” may be misleading as it is difficult to determine using activity measurements alone. A more appropriate term would be “activity offset”.
4. P. 17, lines 27-28: “This is not true for Fourier phases, which exhibit very distinct concentration points and dispersion at slow frequencies.” The authors should clarify what is meant by “slow” frequencies.
5. A further limitation of the study is the imbalance of sex and the fact that it makes controlling for these differences difficult. Although given that the number of subjects is low and

thus further controlling for this in the statistical model is challenging, this fact should still be mentioned.

Minor:

There are instances of grammar and spelling issues throughout the manuscript which hinder clarity for the reader. However, these are manageable through proof-reading of the final manuscript. I have highlighted a few below as well as some other minor comments and questions.

1. In general use of the serial comma would make sentences clearer. For example, "circadian, ultradian, and infradian" as the authors are treating these as separate phenomena.
2. P. 11, line 46, duplication of word "especially specially)
3. P. 12, line 8 "physiologycal"
4. P. 12, line 9, "... at several time-scales", It would be easier to follow if "several" was more specifically defined or replaced with "time-scales of different lengths", based on the references given here. However, the authors could define what is meant by circa-, ultra-, and infra-, in this context.
5. P. 12, line 17 "has offered"
6. P. 12, line 33, it is unclear what is meant by "cycles at every level". Could the authors expand upon this?
7. P. 13, line 7-8, "Moreover, the consideration of Fourier phases in approximately.." should be "approximating"
8. P. 17, lines 29-35: This sentence needs to be rewritten and parsed as it is very difficult to follow.

Decision letter (RSOS-210463.R1)

Dear Dr Rivera

On behalf of the Editors, we are pleased to inform you that your Manuscript RSOS-210463.R1 "A non-parametric model-free analysis of actigraphic recordings of acute insomnia patients" has been accepted for publication in Royal Society Open Science subject to minor revision in accordance with the referees' reports. Please find the referees' comments along with any feedback from the Editors below my signature.

Please submit your revised manuscript and required files (see below) no later than 7 days from today's (ie 17-Nov-2021) date. Note: the ScholarOne system will 'lock' if submission of the revision is attempted 7 or more days after the deadline. If you do not think you will be able to meet this deadline please contact the editorial office immediately.

Please note article processing charges apply to papers accepted for publication in Royal Society Open Science (<https://royalsocietypublishing.org/rsos/charges>). Charges will also apply to papers transferred to the journal from other Royal Society Publishing journals, as well as papers submitted as part of our collaboration with the Royal Society of Chemistry

(<https://royalsocietypublishing.org/rsos/chemistry>). Fee waivers are available but must be requested when you submit your revision (<https://royalsocietypublishing.org/rsos/waivers>).

on behalf of Prof Mark Chaplain (Subject Editor)
openscience@royalsociety.org

Associate Editor Comments to Author:

As well as addressing the remaining referee commentary in your revision, please ensure that you not only provide any electronic supplementary material with the revision but also the 'for review' link for your Dryad deposition (we note you've provided the link that will be 'live' on publication, but this is not accessible to reviewers and editors).

Reviewer comments to Author:

Reviewer: 2

Comments to the Author(s)

The authors have addressed all my comments satisfactorily.

I just have a minor comment.

It would be useful for the authors to review the manuscript for language and sentence construction. Some sentences for example lines 29-35 are too long and ones loses a sense of what is intended. It would be good to shorten the very long sentences.

Reviewer: 3

Comments to the Author(s)

Summary:

In the following manuscript, Marin-Garcia et al, present an alternative methodological approach to analyzing actigraphy recordings from acute insomnia patients. They compare parametric and non-parametric approaches and detail the pros and cons of each. They then present a non-parametric method to analyze these signals using a purely data-driven approach, which is meant to alleviate problems associated with adhering to increases in the number of pre-determined constants. They conclude that the "acrophase" component of these analyses can provide new insight into both circadian and ultradian phenomena.

Comments:

1. Motion probability comparisons: Figure 1 (p.16): It is unclear to me what conclusions the reader should draw from panel A and B. Furthermore, why is the data distribution not given for each group? (I realize that individual values are in panels C and D). Is it possible for the authors to also plot the standard error? Or does it make interpretation difficult due to the range of the data? Regarding the motion probability estimate, the authors states that either 7 or 14 whole days were used (p. 14, line 23). What was the reason for this? Also was any difference seen when averaging the data for these individuals? Furthermore, is this related to the number of male and female subjects in the control group?
2. Related to question one, what was the rationale for taking 1 min values? Were there any attempts to look at larger or smaller (I realize that depending on the recording device this may not be possible) time-scales? Given that the 1 min activity bout is a slightly arbitrary block of time

(and may just be the smallest interval outputted by the device), do the authors think their results would change if data was summed over larger intervals as certain “discrete or stochastic” events might then contribute less?

3. The use of the term “sleep-onset” may be misleading as it is difficult to determine using activity measurements alone. A more appropriate term would be “activity offset”.

4. P. 17, lines 27-28: “This is not true for Fourier phases, which exhibit very distinct concentration points and dispersion at slow frequencies.” The authors should clarify what is meant by “slow” frequencies.

5. A further limitation of the study is the imbalance of sex and the fact that it makes controlling for these differences difficult. Although given that the number of subjects is low and thus further controlling for this in the statistical model is challenging, this fact should still be mentioned.

Minor:

There are instances of grammar and spelling issues throughout the manuscript which hinder clarity for the reader. However, these are manageable through proof-reading of the final manuscript. I have highlighted a few below as well as some other minor comments and questions.

1. In general use of the serial comma would make sentences clearer. For example, “circadian, ultradian, and infradian” as the authors are treating these as separate phenomena.

2. P. 11, line 46, duplication of word “especially specially)

3. P. 12, line 8 “physiologycal”

4. P. 12, line 9, “... at several time-scales”, It would be easier to follow if “several” was more specifically defined or replaced with “time-scales of different lengths”, based on the references given here. However, the authors could define what is meant by circa-, ultra-, and infra-, in this context.

5. P. 12, line 17 “has offered”

6. P. 12, line 33, it is unclear what is meant by “cycles at every level”. Could the authors expand upon this?

7. P. 13, line 7-8, “Moreover, the consideration of Fourier phases in approximately..” should be “approximating”

8. P. 17, lines 29-35: This sentence needs to be rewritten and parsed as it is very difficult to follow.

===PREPARING YOUR MANUSCRIPT===

one version should clearly identify all the changes that have been made (for instance, in coloured highlight, in bold text, or tracked changes);

While not essential, it will speed up the preparation of your manuscript proof if you format your references/bibliography in Vancouver style (please see

<https://royalsociety.org/journals/authors/author-guidelines/#formatting>). You should include DOIs for as many of the references as possible.

===PREPARING YOUR REVISION IN SCHOLARONE===

<https://royalsociety.org/journals/authors/author-guidelines/#data>. You should ensure that you cite the dataset in your reference list. If you have deposited data etc in the Dryad repository,

please only include the 'For publication' link at this stage. You should remove the 'For review' link.

-- If you are requesting an article processing charge waiver, you must select the relevant waiver option (if requesting a discretionary waiver, the form should have been uploaded, see 'File upload' above).

-- If you have uploaded any electronic supplementary (ESM) files, please ensure you follow the guidance at <https://royalsociety.org/journals/authors/author-guidelines/#supplementary-material> to include a suitable title and informative caption. An example of appropriate titling and captioning may be found at https://figshare.com/articles/Table_S2_from_Is_there_a_trade-off_between_peak_performance_and_performance_breadth_across_temperatures_for_aerobic_scope_in_teleost_fishes_/3843624.

Author's Response to Decision Letter for (RSOS-210463.R1)

See Appendix B.

Decision letter (RSOS-210463.R2)

Dear Dr Rivera,

I am pleased to inform you that your manuscript entitled "A non-parametric model-free analysis of actigraphic recordings of acute insomnia patients" is now accepted for publication in Royal Society Open Science.

The proof of your paper will be available for review using the Royal Society online proofing system and you will receive details of how to access this in the near future from our production office (openscience_proofs@royalsociety.org). We aim to maintain rapid times to publication after acceptance of your manuscript and we would ask you to please contact both the production office and editorial office if you are likely to be away from e-mail contact to minimise delays to

publication. If you are going to be away, please nominate a co-author (if available) to manage the proofing process, and ensure they are copied into your email to the journal.

on behalf of Professor Mark Chaplain (Subject Editor)
openscience@royalsociety.org

Appendix A

Dear Editor:

In behalf of the authors of the research article entitled “A non-parametric model-free analysis of actigraphic recordings of acute insomnia patients”, I submit now the new revised manuscript. This version includes the answer to all the referee’s comments. We believe the paper became clearer and more interesting due to these comments and we thank you most sincerely.

Enclosed you will find the clean revised version of the manuscript, a marked-up version with all the changes made from the previous article file, and all the figures.

Sincerely yours,

Dr. Ana Leonor Rivera

Reviewer: 1

Comments to the Author(s)

This manuscript considers analysis of time series actigraph data comparing recordings from control and insomniac groups using Fourier decomposition of activity counts to assess differences in the two populations. My comments fall into two areas 1) Methodology/results and 2) Interpretation.

1. Methodology.

The rationale for using non parametric methods is discussed in the manuscript, the motivation given appears reasonable. In section 2 the techniques are presented, this section could benefit from better integration between the methods, particularly the assumptions being made of the data, regarding stationarity for Fourier decomposition, presence (or otherwise) of stochastic components, and the rationale for using the Jaccard method and K-S test for comparing the activity counts and frequency analyses, respectively.

Thanks for your comment about methodology. We hope that in the new version of the manuscript a better integration of the methods and results is presented.

Regarding stationarity and stochasticity, a modification is included when describing the datasets in the last paragraph of subsection 2.a) (page 3):

“It is worth noting that although these kind of datasets are not stationary in the broad sense of the term, they are approximately periodic, meaning that leaving extraordinary events aside, subjects exhibit stable cycles of activity throughout the observation period. This by no means exclude the presence of stochastic components in the observed signals, and it is precisely because the ignorance about true nature of the underlying noise that non-parametric methods are preferred. Previous studies analyzed actigraphic time series using the spectral approach [19,20,23,24,26,29,30,54-61]; however, in all cases the emphasis is placed on the relevance of the *amplitudes* of each spectral component, or interval of components, disregarding the complementary information contained in the corresponding *phases*.”

This argues that given the subject-specific variable sources of noise within the biological signal, any statistical model is problematic to present since no behavioral data for each subject is available. However, it is expected that within the restrictions imposed to this study, useful results can be obtained to describe group-wise differences, this with the assumption that subject-specific characteristics will be less pronounced when normalized in a group-wise manner.

Following this rationale, another modification is included in the final paragraph of subsection 2.c):

“KS is often used to test for statistical equality between small samples with good results [64]. It is sensitive to both the scale (or spread) and location (or center), and requires no parameters to be specified [65]. These features are desirable with the problem at hand, since we wish to test for equality between samples which ranges vary in orders of magnitude such as Fourier

amplitudes, and samples that are expected to vary in location of the accumulation point such as Fourier phases, neither of which are expected to be of Gaussian origin. Other statistical tests for the case of normality test perform much better [66], but KS does not require Gaussian data.”

On the grounds that since no model of noise is to be constructed, a non-parametric significance test for both Fourier amplitudes and phases between groups should be sensitive for both location (of the average/median) and scale (of the ranges of dispersion), the KS statistic fulfills these requirements.

For the use of the Jaccard distance, the rationale centers in its simplicity, ease of interpretation and computational efficiency. The starting paragraph of section 2.d).(ii) (page 5) describes in a more robust manner the background and usage of the Jaccard distance:

“Using the normalization described in the previous section, a non-parametric comparison between PDFs is used to estimate the distance between them. The Jaccard distance metric is a straightforward, simple and computationally efficient method to estimate the distance between probability distributions, is closely related to many other distance metrics between PDFs [67], and it offers a intuitive interpretation of distance between pairs of PDFs. Based in its geometric counterpart: the Jaccard Index [68], it has been since used in the form of a metric between PDFs in artificial intelligence [69], chemistry [70], bioinformatics [71,72], and generalizations to higher dimensions and multivariate systems [73].”

The results in figure 1, and equations (4) and (5) present the data as PDF values. However, these are just mean activity counts for a single subject in (4) and across each group in (5). The use of mean activity counts seems reasonable, but presenting these as PDFs is not justified here, there is no discussion of normalisation, PDF estimation or what this is a probability of? Presenting these as measures of activity counts would be more appropriate in this case.

We change the methodology taking in count your comment. Subsection 2.d).(i) (page 4) has been re-written:

“Using the time-resolved activity counts $x_s(m)$ of a subject s observed D_s whole days (either 7 or 14), probability density functions (PDFs) of motion as a function of time (in minutes) at 24hr time-scale for both subject- and group-wise levels are constructed. In this case, the 24hr. activity counts by minute normalized by the total activity counts for that day yields a PDF given that the summation over all resulting values is 1. To obtain these PDFs, each $x_s(m)$ is first partitioned into D_s 24hr segments: $y_{s,k}(t)$, with t referring to each minute and k to each day ($t=1 \dots 1440$, and $k=1 \dots D_s$), hence, to construct all the 24hr subject-wise motion probability density functions, corresponding values at each minute are normalized with the following: ...”

Thus, the normalization of motion activity counts as a fraction of total activity counts per day yields a PDF in the sense that the corresponding values can be treated as probability of activity as a function of time (in minutes) since the summation of these values over all minutes in a day is equivalent to 1, then, the group-wise averaging of motion probabilities per subject also yields a PDF. It would be equivalent to speak of motion averages in the group-wise manner, but since the interpretation of the normalization and averaging opens up the possibility of using the Jaccard distance as distance measure between group-wise PDF the latter interpretation is proposed.

We also change the caption of figure 1, now it reads:

Motion Probability Estimates. 24hr. probability of motion as a function of time for one control (a) and one insomniac subject (b). 24hr. probability of motion for all control (c) and all insomniac subjects (d), mean values over all subjects and all days are plotted with solid black curves. (e) Mean values of the 24hr. probability of motion for control (dashed blue curve) and insomniac (solid red curve) groups. (f) Jaccard distance between mean values of the 24hr. probability of motion for control and insomniac groups as a function of temporal displacement; inset shows in detail the behavior within the interval [0-150] min.

Some additional clarity on what is presented in Figure 2 would be useful. Panels a), b) show individual log periodograms, with the average periodogram overlaid. Panel e) also contains some form of average spectral estimate, but the values are different. What is the relationship between these two sets of spectral estimates?

We change caption of Figure 2:

“Spectral Estimates. Fourier amplitudes for control (a) and insomniac groups (b) in log-log scale, mean values at each spectral component (k) are plotted with solid curves. Fourier phases for control (c) and insomniac (d) groups in semi-log scale on the horizontal axis, mean values at each spectral component (k) are plotted with black circles (control group) and black diamonds (insomniac group). (e) Mean values of Fourier amplitudes in log-log scale for control (blue dashed curve) and insomniac (solid red curve), respectively. (f) Mean values of Fourier phases in semi-log scale on the horizontal axis for control (blue circles) and insomniac (red diamonds), respectively. Significant ($p < 10E-2$) phase differences found using the Kolmogorov-Smirnoff test are indicated by a red rectangle. (g) and h) results of the application of the non-parametric Kolmogorov-Smirnoff test between control and insomniac groups for Fourier amplitudes and phases at each spectral component (k), respectively; both panels are shown in log-log scale; significant ($p < 10E-2$) phase differences in slow components found using the Kolmogorov-Smirnoff test are indicated also by a red rectangle.”

The panels a) and b) of figure 2 do refer to individual log periodograms for control and insomniac groups, respectively, a wide range of values can be observed for both groups at every spectral component, we choose to emphasize this with the scale used, and to also provide with a plot of the mean values only presented with red and blue curves, respectively, in panels e) and f).

It would be useful to include recognition that these are just estimates, a clearer statement, possibly with equation, on how the averages across subject were calculated could be included.

In the new version of the manuscript a clearer description of normalisations is included in subsection 2.d).(i) (page 4). Name of the figures state now that are “Spectral Estimates”.

Following up the Bloomfield reference, some indication of confidence limits on these log spectra plots would be a useful addition. This would give some context for the 4-5 orders of magnitude we see in the ranges of values for individual subjects.

We hope that this has been addressed with the modifications in paragraphs in section 2.

2. Interpretation of results is weak in the context of comparing the two groups. The discussion contains 3 numbered points.

1) Phase – this just says phase is a useful measure, but makes no attempt at inference. What interpretation should the reader give to the phase difference highlighted in Fig 2f) and that sometimes the phase is greater for insomniac and sometimes greater for control. Thus, what does a phase of $\pi/2$ signify for index $k=1$?

Specific values of phases at the first spectral component reveal the sleep-onset delay between groups. This explained by phase differences (in minutes) of component 1. Specific values of Fourier phases at this component indicate the relationship between the (circadian) light-dark cycle and the activity profiles of each subject and group.

2) The point on statistical tests should include more details of where the significant differences were found and the point regarding the need for additional analysis should contain more details of how this can be followed up.

We hope to answer this concern with the inclusions in section 2 along with the next paragraph at the end of section 3.b) (page 7):

“Because the experimental design, data acquisition protocols, including the length of observation periods, sampling frequency, number of participants, and unrestricted activity logging, information regarding the range of dispersion for either activity counts and all derived measures cannot be ascribed to specific noise sources, however, since weak periodicity can be expected from a large and diverse subject pool, both central values and dispersion ranges are expected to contain useful information when stablishing archetypical profiles for each group, this is true for the estimation of motion activity PDFs, as well as Fourier amplitudes and phases. Considering the fast-paced rhythm of activity observed in humans, both in healthy and unhealthy human populations [20,27,76], the main sources of stochasticity within the data are expected to be present at corresponding fast spectral components. Previous investigations inform the expectation of finding a sleep--onset delay of ~ 90 min. between groups [36], represented by equivalent phase difference at the first spectral component $\phi(1)_C$ and $\phi(1)_I$ for the control and insomniac groups, respectively. To support the use of these differences, test for statistical equality between samples are first performed. Panels g) and h) of figure 2 present the (\log_{10})p-values of the KS--test for Fourier amplitudes and phases, respectively, and for

all spectral components. Significance tests show that Fourier amplitudes show only marginal significance ($10^{-2} < p < 10^{-1}$) for spectral components ($k=17,19,20$), whereas phases are found to be significantly different ($p < 10^{-2}$) between control and insomniac groups in slow spectral components ($k=1,2,4,5,7$). The difference in amplitude between groups corresponding to spectral components also called BRAC cycles [53] is also found through parametric methods using the same dataset [36]. In this case, for the ~ 85 , ~ 75 , and ~ 72 min. periodicities (components ($k=17,19,20$), respectively), significant differences in Fourier amplitudes is found as well. Notably, when transformed into minutes, phase differences between central values at the spectral components with a significant phase difference do correlate with results from previous parametric investigations, expressing a significant phase difference of 20 degrees at the 1440min. spectral component ($\phi(1)$), which translates to 80 minutes when expressed in terms of that cycle. This is within the plateau found for the Jaccard distance between group-wise activity profiles in panel f) of figure \ref{fig1}. Using the available information, it is unclear the meaning of significance of phase differences found in other spectral components, namely $\phi(2)$, $\phi(4)$, $\phi(5)$, and $\phi(7)$ since only for the 24hr. cycle ($\phi(1)$) can be intuitively related to physiological changes [77].”

In particular, regarding components which are found statistically different between groups, amplitudes are found to correlate to results obtained with the same datasets but with parametric methods, meanwhile phases corroborate the sleep-onset delay between group-wise PDFs.

3) This contribution is largely speculative and makes no reference to any of the results presented. For example, what do the results in Figs 1 and 2 tell us about synchronisation of circadian rhythms or neurogenesis? This needs clarification, or possibly to be removed for some more concrete discussion. A key point in the analysis is the 90 minute delay detected using the Jaccard analysis, but this is not mentioned in the discussion. What is the significance of this delay for our understanding and comparing the two groups?

With the modifications in previous paragraphs, we hope we have addressed this. Moreover, all of section 4 has been modified to better reflect both the integration between methods and the clearer statements about the datasets as well as the results. The three last paragraph of section 4 (page 9) we hope provide with a more robust arguments for the connections we establish between multioscillatory phenomena, statistical analyses, biological time-series, and non-parametric methods.

Minor

1. Abstract: “concepts such as acrophase are staple in chronobiology”. Clarify meaning of this.

We modify the abstract now it reads:

“Both parametric and non-parametric approaches to time-series analysis have advantages and drawbacks. Parametric methods, although powerful and widely used, can yield inconsistent

results due to the oversimplification of the observed phenomena. They require the setting of arbitrary constants for their creation and refinement, and, although these constants relate to assumptions about the observed systems, it can lead to erroneous results when treating a very complex problem with a sizable list of unknowns. Their non-parametric counterparts, instead, are more widely applicable but present a higher detrimental sensitivity to noise and low density in the data. For the case of approximately periodic phenomena, such as human actigraphic time-series, parametric methods and concepts such as *acrophase* are widely used in chronobiology, especially specially when studying healthy and diseased human populations. In this work, we present a non-parametric method of analysis of actigraphic time-series from insomniac patients and healthy age-matched controls, the method is fully data-driven, reproduces previous results in the context of sleep-onset delay and, crucially, extends the concept of *acrophase* not only to *circadian* but also for *ultradian* spectral components.”

2. In equation (2) the time index from be 0 to (1-1) for a DFT definition.

Changed.

3. In Line 128 what is “proper probability”.

It is changed, it refers to “probability density functions”.

4. Lines 187-189 incorrect figure references.

Now the reference to figure is double-checked and corrected.

5. Line 261, wavelet vs Fourier these criticisms also apply to the DFT where frequency resolution depends on sequence length and any data windowing.

Text in section 2 is modified.

Reviewer: 2

Comments to the Author(s)

This manuscript describes a non-parametric method of analysis of actigraphic time series from 12 insomniac patients and healthy age-matched controls, that is fully data-driven.

Detailed comments are below:

On Page 4, Line 59, the authors state that their technique avoids the difficulties of making assumptions about noise and true signal. But this means the outcome of the analyses provide no insights on whether the results reflect noise or signal. How is this resolved?

With the modifications in section 2 and 3, which include the expected strong of periodicity and simultaneous high dispersion in activity counts for the diverse population under observation, only group-wise profiles are to be compared between groups, as well as finding only high significance in difference between samples is found at slow spectral components, highlighting that the noise-ratio at those spectral components becomes highly detrimental with the experimental set up for this investigation.

On Page 9, Line 166 the authors discuss results from two subjects to highlight differences between healthy sleepers and insomniacs. How are these findings different from that of a traditional approach?

In this case, temporal shifts and changes in activity patterns are detected through non—parametric means, specifically, using normalizations that enable the interpretation of activity counts as a PDF, thus providing with the possibility of use of non-parametric, geometric-inspired measures of distance between group-wise estimated PDFs. We hope this has been presented in a clearer manner with the modifications in section 2.

In page 15, Line 252, the authors state ' we assessed motion activity counts with non-parametric, model-free methods such that no assumption is made regarding the origins of the data; such assumptions are particularly problematic in diseased populations with less pronounced circadian rhythms.' Why is that a problem in individuals with less pronounced circadian rhythms? Please explain with specific reasons.

In severe pathologies, such as Alzheimer's disease, the circadian rhythm can be completely absent. Parametric methods that presuppose an underlying model, such as cosinor analysis, are not applicable in these cases. Also in less severe pathologies, such as insomnia, a user-defined model may not correspond to the actual patterns present in the data. Given the expected diversity between subjects, as well as the diverse range of activity profiles each subject can express, and that insomnia alters activity patters in several ways, the absence of models for noise or stochasticity would make interpretations about centrality and dispersion alone should be disregarded, instead, we propose the use of non-parametric tests for equality that are sensitive to changes in both centrality and dispersion, such as the KS test.

On page 16, line 265 the authors state 'each spectral component can be determined and associated with specific physiological and environmental factors, such as light-darkness, melatonin-onset, and other cycles, physiological or psychological. It would be useful to highlight these associations with the results of your analyses. Also what psychological cycles are you referring to?'

The new version of the manuscript offers a more comprehensible interpretation of the presented results. In this case, only for the phase difference in the case of the first spectral component can be interpreted as sleep-onset delay, relating this to the sleep-onset estimated using the Jaccard Distance metric in figure 1.

The authors have provided a new analytic technique for actigraphy data. It would be useful if they could substantiate with specific, explicit results their statements in the discussion and conclusions. For example in their conclusions the authors state 'that spectral and statistical properties for infradian, circadian and ultradian rhythms in human can be studied through continuous actigraphic recordings and using non-parametric approaches.' Please provide specific examples with your data analyses. Secondly, please discuss in detail the environmental cycles to which the results of your analyses can be anchored.

We hope the new version of the manuscript better reflect the usefulness of the proposed methodology. Additions to the methodology and results sections have been included to better describe the data normalizations and estimations, and, crucially, relationships between observed phase differences in the Fourier sense and the estimated acrophase reported with previous parametric methods, interpretations and conclusions have been rewritten to also include these considerations, chiefly, modifications in subsection 2 and 3, as well as final paragraphs in section 4.

Appendix B

Comments to the Author(s)

The authors have addressed all my comments satisfactorily.

I just have a minor comment: It would be useful for the authors to review the manuscript for language and sentence construction. Some sentences for example lines 29-35 are too long and ones loses a sense of what is intended. It would be good to shorten the very long sentences.

In all the text we shorten the very long sentences. In particular, this specific paragraph has been rewritten, now reads:

Because the experimental design, data acquisition protocols, including the length of observation periods, sampling frequency, number of participants, and unrestricted activity logging, information regarding the range of dispersion for either activity counts and all derived measures cannot be ascribed to specific noise sources. However, since weak periodicity can be expected from a large and diverse subject pool, both central values and dispersion ranges are expected to contain useful information when establishing archetypal profiles for each group, This is true for the estimation of motion activity PDFs, as well as Fourier amplitudes and phases.

Comments to the Author(s)

Summary:

In the following manuscript, Marin-Garcia et al, present an alternative methodological approach to analyzing actigraphy recordings from acute insomnia patients. They compare parametric and non-parametric approaches and detail the pros and cons of each. They then present a non-parametric method to analyze these signals using a purely data-driven approach, which is meant to alleviate problems associated with adhering to increases in the number of pre-determined constants. They conclude that the “acrophase” component of these analyses can provide new insight into both circadian and ultradian phenomena.

Comments:

1. Motion probability comparisons: Figure 1 (p.16): It is unclear to me what conclusions the reader should draw from panel A and B. Furthermore, why is the data distribution not given for each group? (I realize that individual values are in panels C and D). Is it possible for the authors to also plot the standard error? Or does it make interpretation difficult due to the range of the data? Regarding the motion probability estimate, the authors states that either 7 or 14 whole days were used (p. 14, line 23). What was the reason for this? Also was any difference seen when averaging the data for these individuals? Furthermore, is this related to the number of male and female subjects in the control group?

Thank you for pointing this, it needs clarification.

Panels A and B present individual PDFs from two subjects, one for each group. Panels C and D present all values for all subjects in circles and diamonds, whereas solid black curves show the central values at each minute of the 24hr cycle, also for each group. We wished to present only the central value in overlay since also presenting dispersion ranges would obscure the main idea of a high dispersion being present at the motion count level in both groups.

Full weeks were chosen in order to minimize the sub--representation of few subjects present in the raw datasets, but also, we wished to include week--long cycles when the dataset allowed for it. This is not connected to the sex of any participant.

2. Related to question one, what was the rationale for taking 1 min values? Were there any attempts to look at larger or smaller (I realize that depending on the recording device this may not be possible) time-scales? Given that the 1 min activity bout is a slightly arbitrary block of time (and may just be the smallest interval outputted by the device), do the authors think their results would change if data was summed over larger intervals as certain “discrete or stochastic” events might then contribute less?

This is solely because of choices made during the experimental protocol design, as well as the hardware and software used to obtain the measurements of activity counts. Yes, alternative parametric configurations may be used, we expect that this coarse--graining of the data retains descriptive power to an extent, but after a “maximum” summation length all profiles should present no significant differences.

3. The use of the term “sleep-onset” may be misleading as it is difficult to determine using activity measurements alone. A more appropriate term would be “activity offset”.

Agreed, in order to address it as sleep-onset, clinical validation should be first established between activity counts and sleep. It is changed the term “sleep-onset” on all the text to “activity offset”.

4. P. 17, lines 27-28: “This is not true for Fourier phases, which exhibit very distinct concentration points and dispersion at slow frequencies.” The authors should clarify what is meant by “slow” frequencies.

Thank you for pointing this. A final paragraph at the end of section 2.b. has been added, we hope this helps to clarify the meaning of these usage of fast and slow. The paragraph reads:

With the chosen observation length period, the first spectral component describes the amplitude and phase of the circadian (≈ 24 hr) cycle, the slowest cycle using the observation length of 24hr. Strictly speaking, all other (faster) spectral components can be described as ultradian (> 24 hr) components. However, it is yet unclear which ultradian spectral components can be meaningfully ascribed to biological cycles. Given the restrictions by the experimental design and data acquisition protocols, infradian (< 24 hr) spectral components are not studied in this work since datasets are normalized to a 24hr time length for all subjects in both groups.

5. A further limitation of the study is the imbalance of sex and the fact that it makes controlling for these differences difficult. Although given that the number of subjects is low and thus further controlling for this in the statistical model is challenging, this fact should still be mentioned.

Thank you for pointing to this. Text has been updated. First line of second paragraph of section 4 now reads:

All of this comes with limitations due to the experimental design, data acquisition protocols, observation period length, sampling density, and the sizes of each group. These limitations prevented us for controlling for age, sex, and socio-economic status, variables which certainly play an important role in this disease as well as in chronobiology in general.

Minor:

There are instances of grammar and spelling issues throughout the manuscript which hinder clarity for the reader. However, these are manageable through proof-reading of the final manuscript. I have highlighted a few below as well as some other minor comments and questions.

1. In general use of the serial comma would make sentences clearer. For example, “circadian, ultradian, and infradian” as the authors are treating these as separate phenomena.

Text has been updated.

2. P. 11, line 46, duplication of word “especially specially)

Text has been updated.

3. P. 12, line 8 “physiological”

Text has been updated.

4. P. 12, line 9, “... at several time-scales”, It would be easier to follow if “several” was more specifically defined or replaced with “time-scales of different lengths”, based on the references given here. However, the authors could define what is meant by circa-, ultra-, and infra-, in this context.

Text has been updated.

5. P. 12, line 17 “has offered”

Text has been updated.

6. P. 12, line 33, it is unclear what is meant by “cycles at every level”. Could the authors expand upon this?

We agree this needs clarification. We hope the additions to the third paragraph of this section clarify this, now it reads:

In this context, and given the multioscillatory nature of physiological [2,43-49], and psychological [1,50-53] cycles at the molecular, tissue, and system levels. It would be reasonable to extend the concept of acrophase to specific spectral components or intervals which have been shown to have physiological or psychological relevance. Classically, acrophase has been only applied to circadian ($\approx 24\text{hr}$) cycles, whereas ultradian ($> 24\text{hr}$), and infradian ($< 24\text{hr}$) cycles are not subject to characterization using this concept.

7. P. 13, line 7-8, “Moreover, the consideration of Fourier phases in approximately..” should be “approximating”

Text has been updated.

8. P. 17, lines 29-35: This sentence needs to be rewritten and parsed as it is very difficult to follow.

We appreciate the feedback on this specific paragraph, it has been rewritten, now reads:

Because the experimental design, data acquisition protocols, including the length of observation periods, sampling frequency, number of participants, and unrestricted activity logging, information regarding the range of dispersion for either activity counts and all derived measures cannot be ascribed to specific noise sources. However, since weak periodicity can be expected from a large and diverse subject pool, both central values and dispersion ranges are

expected to contain useful information when establishing archetypal profiles for each group, This is true for the estimation of motion activity PDFs, as well as Fourier amplitudes and phases.